# Comparative evaluation of five rapid PCR platforms for respiratory virus detection

Pieter W. Smit[1,2], Peter de Man[1], Henk Brand[1], Simone Breijer[1], Mariska van Wijk[1], Adam Meijer[3]*, David S. Y. Ong[1,4]*

1 Department of Medical Microbiology and Infection Control, Franciscus Hospital, Rotterdam, the Netherlands, 2 Molecular Diagnostics Unit, LMM, MaasstadLab, Maasstad Hospital, Rotterdam, the Netherlands, 3 Centre for Infectious Disease Control, National Institute for Public Health and the Environment, Bilthoven, the Netherlands, 4 Department of Epidemiology, Julius Center for Health Sciences and Primary Care, University Medical Center Utrecht, Utrecht, the Netherlands

* adam.Meijer@RIVM.nl (AM); davidsyong@gmail.com (DSYO)

## Abstract

### Background

The demand for rapid molecular diagnostics for respiratory viruses has increased substantially. Several point-of-care PCR platforms have become available, yet comparative performance data remain limited.

### Objectives

To evaluate the diagnostic accuracy and operational reliability of four rapid PCR platforms for detection of severe acute respiratory syndrome coronavirus 2 (SARS-CoV-2), influenza A and B viruses (IAV, IBV), and respiratory syncytial virus (RSV), in comparison with the GeneXpert (Cepheid, USA) platform.

### Methods

Nasopharyngeal swabs from patients with respiratory symptoms were tested using the GeneXpert, positive samples were subsequently analysed on four alternative systems: the 30-minute and 1-hour M10 (SD Biosensor, South Korea) assays, Flash-Detect™ Flash10 (Coyote Bioscience, China), Vivalytic (Bosch Healthcare Solutions, Germany), and Galaxy Lite (Igenesis, China). Additional lower viral load samples and cultured IAV/ IBV strains were included.

### Results

A total of 223 GeneXpert positive samples were prospectively analysed. Flash10 showed 94.6% overall agreement, missing SARS-CoV-2 (n = 4, GeneXpert Cycle threshold (Ct) min-max; 37.7–42.2), IAV (n = 5, 32.8–37.7), and IBV (n = 3, 26.5–36.7). Vivalytic showed 83.0% overall agreement, missing SARS-CoV-2 (n = 16, 30.4–42.2),

**Data availability statement:** All relevant data are within the manuscript and its Supporting information files.

**Funding:** The author(s) received no specific funding for this work.

**Competing interests:** The authors have declared that no competing interests exist.

IAV (n = 9, 26.8–37.7), IBV (n = 9, 27.2–36.7), and RSV (n = 4, 31.5–37.0). Galaxy Lite achieved 88.2% overall agreement but failed in 27.2% of test runs. With a smaller sample size the M10 (30-minute) assay showed 98.6% overall agreement with GeneXpert, missing one SARS-CoV-2 case (Ct 39.7).

## Conclusion

Among four platforms, the M10 (30-min version) and Flash10 platforms demonstrated the highest agreement rates with the GeneXpert. The variability in performance highlights the importance of independent platform evaluation.

## Introduction

Respiratory viral infections, particularly those caused by severe acute respiratory syndrome coronavirus 2 (SARS-CoV-2), influenza A and B viruses (IAV and IBV), and respiratory syncytial virus (RSV), continue to impose a significant global health burden. These pathogens are associated with considerable morbidity and mortality, placing immense pressure on healthcare systems and underscoring the need for rapid and reliable diagnostic strategies [1–3]. Timely identification of these infections is critical not only for initiating appropriate clinical management but also for implementing effective infection control measures and public health interventions [2–4].

Although laboratory-based reverse transcription polymerase chain reaction (RT-PCR) assays have long been regarded as the diagnostic gold standard, their reliance on centralized laboratory infrastructure and extended turnaround times can delay critical decision-making [2]. In response to these challenges, there has been a notable shift toward the development and deployment of point-of-care (POC) molecular diagnostics. These rapid PCR platforms are designed to deliver accurate results within a significantly reduced timeframe outside of a laboratory setting, facilitating prompt treatment decisions and improving patient management in both hospital and community settings [5–7].

Among the available POC solutions, the GeneXpert system (Cepheid, USA) has established itself as a robust reference platform for the simultaneous detection of multiple respiratory pathogens. However, the diagnostic landscape is rapidly evolving, with several new platforms entering the market that claim comparable or superior performance [6–8]. Despite this progress, comprehensive head-to-head evaluations comparing a multitude of platforms remain scarce, highlighting the need for comparative studies assessing their analytical performance, sensitivity, specificity, and operational feasibility. Additionally, the economic burden associated with large-scale testing has grown substantially due to more frequently testing for respiratory viruses since the emergence of coronavirus disease 2019 (COVID-19) and greater recognition of the importance of detection for not only SARS-CoV-2 and IAV/IBV but also RSV in older adults and those with pulmonary or cardiac comorbidities [9]. The high costs associated with established platforms such as GeneXpert necessitate the exploration of cost-effective alternatives without compromising diagnostic accuracy.

In this study, we compared four rapid PCR platforms—M10 (SD Biosensor, South Korea), Vivalytic (Bosch, Germany), FlashDetect™ Flash10 (Coyote Bioscience, China), and Galaxy Lite (Igenesis, China)—against the GeneXpert (Cepheid, USA) as the reference standard for the detection of SARS-CoV-2, IAV/IBV, and RSV. The M10 and Vivalytic platforms have been available for a few years, while the FlashDetect™ Flash10 and Galaxy Lite were introduced in 2024. Our primary objective was to evaluate these platforms in terms of their diagnostic accuracy, usability, and operational costs to inform clinical and laboratory practices.

## Materials and methods

### Study design and setting

This prospective study was conducted at the Franciscus Hospital between November 2024 and March 2025. Patients presenting with respiratory symptoms were consecutively tested for SARS-CoV-2, IAV/IBV, and RSV using Xpert Xpress SARS-CoV-2/Flu/RSV Plus assay on the GeneXpert platform (Cepheid, USA) as part of routine diagnostic care. Obtained nasopharyngeal swabs (MANTACC, Miraclean technology Co, China) were placed in Viral Transport Medium (HiViral Transport Medium, HiMedia Laboratories, India). No specific inclusion criteria regarding patient characteristics were applied, ensuring a broad and representative sample of the patient population. Testing was performed as part of the triage protocol for patients presenting respiratory symptoms at the emergency department. Sample size calculation was performed with the aim to be able to detect at least detect a 15% difference in agreement per viral target with 80% power and an α of 0.05 (McNemar), which showed that 53 samples were required per viral target.

To ensure adequate representation of each targeted virus, at least 50 positive samples for respiratory virus were prospectively included. Positive samples according to GeneXpert were enrolled consecutively until the target number per virus was reached. No separate group of negative samples was included. Single-target positives were considered negatives for the other virus targets. A small number of co-infections were observed during routine testing; these were included based on the dominant target for which they were selected. An additional group of 30 IAV positive samples with GeneXpert Ct values >27 was prospectively collected during the same study period to evaluate samples with lower viral loads. The study aimed to evaluate and compare the diagnostic performance of four rapid PCR platforms—M10 (SD Biosensor), Vivalytic (Bosch), FlashDetect™ Flash10 (Coyote Bioscience), and Galaxy Lite (Igenesis)—with the GeneXpert assay as the reference method. The analyses on the evaluation platforms were mostly performed on the same day as initial testing but with a maximum delay of 48 hours for clinical samples during the weekends. To represent most clinical settings, invalid test results were repeated, but discrepant test results in comparison to other platforms were not repeated.

Midway through the study period, SD Biosensor introduced a new 30-minute assay; Standard M10 FLU/RSV/SARS-CoV-2 FAST, replacing the original 60-minute version. Samples processed after this transition were analyzed using the new 30-minute protocol cartridges with current Research Use Only (RUO) label. This change was documented and considered in the data analysis to assess any diagnostic performance differences between the two versions.

The Institutional Review Board (IRB) approved the study protocol (IRB protocol number 2024−075) and declared that this study does not fall within the scope of the Dutch Medical Research Involving Human Subjects Act.

### Specimen collection and inclusion of cultured influenza virus strains

Nasopharyngeal swabs were collected following standardized procedures and were immediately transported to the microbiology laboratory located within the hospital. Each sample was typically processed within one hour after arrival.

To further assess the platform's capability to detect various influenza virus strains –given the virus's high genetic diversity- a panel of cultured IAV and IBV viruses, representing different genetic clades and H- and N-subtypes of IAV and different genetic clades of B/Victoria lineage viruses were included. This strain set comprised of human seasonal A(H1N1)pdm09 clade 5a.2a (C.1.9.1), A(H3N2) clade 2a.3a.1 (J.2.1, J.2, and J.1.1) and B/Victoria clade V1A.3a.2 (C.5.6,

C.5.1, and C.5.7) lineage viruses of the same period that study clinical samples were collected. This set also contained specimens from a previous carried out External Quality Assessment (EQA) 2023 among clinical diagnostic laboratories in the Netherlands on zoonotic influenza viruses, comprising human seasonal A(H1N1)pdm09 and A(H3N2) strains with mutations in the M-genome segment that caused previously increased false negatives with some commercial assays, and avian (H7N2, H5N6, H5N1, (HA clade 2.3.4.4b in two concentrations)) and swine (H1N1v, H1N2v isolated from human cases) influenza A virus subtypes that previously caused or might cause a zoonotic infection.

## Statistical analysis

Diagnostic performance of each rapid PCR platform was determined by comparing the results to those obtained with the GeneXpert assay. Key performance metrics such as concordance and percentage agreement ratios, including qualitative assessments of user experiences and operational reliability, were assessed for each platform.

## Results

A total of 223 nasopharyngeal samples prospectively were included from patients presenting to the hospital (Table 1). Prospective samples were categorized into target groups based on GeneXpert results: positive for SARS-CoV-2, IAV, IBV, and RSV. The group labeled as "negative" refers to samples that tested negative for the virus in question (e.g., SARS-CoV-2) but may have tested positive for another target. Most platforms demonstrated high reliability in processing samples, generating valid results in nearly all cases. However, the Galaxy Lite platform showed a markedly higher error rate even after replacing the system. Out of the 195 samples tested on this platform, 53 (27.2%) failed to produce a valid result. Due to this high failure rate, Galaxy Lite was excluded from further testing midway through the study.

The agreement of each platform with the GeneXpert reference method is summarized in Table 1. Although fewer samples for M10 (30-minute version) compared to Flash10 were tested, both platforms consistently demonstrated high agreement percentages (Table 1). Vivalytic demonstrated variable performance for prospective samples, with high agreement for RSV (93.5%) and negative samples (100%), but lower concordance for SARS-CoV-2 (72.4%) and IAV (84.7%) compared to the other platforms. The M10 in its earlier 60-minute configuration showed high agreement with the reference method but missing low viral load SARS-CoV-2 and IAV samples, while the updated 30-minute version performing comparably or slightly better across most targets (Table 1).

Among the undetected GeneXpert positive samples in the test platforms, GeneXpert Ct values were generally high, though some platforms failed to detect targets at lower Ct values. Focusing on prospective samples; for M10 (1-hour), missed detections included SARS-CoV-2 (n = 8, GeneXpert Ct range (min-max) 33.0–42.2), IAV (n = 2, 36.7–37.7), and IBV (n = 1, 36.7). The updated 30-minute version of the M10 test only missed one SARS-CoV-2 case (Ct 39.7). Flash10 failed to detect SARS-CoV-2 (n = 4, Ct min-max; 37.7–42.2), IAV (n = 5, 32.8–37.7), and IBV (n = 3, 26.5–36.7). For Galaxy Lite, missed detections included SARS-CoV-2 (n = 10, 32.9–42.2), IAV (n = 3, 29.2–35.3), IBV (n = 4, 14.8–36.7), and RSV (n = 6, 20.3–35.4). Vivalytic showed the highest number of missed detections for SARS-CoV-2 (n = 16, 30.4–42.2), and IAV (n = 9, 26.8–37.7); IBV (n = 9, 27.2–36.7) and RSV (n = 4, 31.5–37.0) samples were also not detected.

All 'false' positives detected by one of the platforms in comparison to GeneXpert were not confirmed by the other platforms and were therefore regarded as true false positives (Table 1).

The reference influenza strains (see material & method section) were detected by all platforms, except for Vivalytic that missed repeatedly H1N1v, a swine flu virus strain isolated from a patient.

From the IAV positive clinical samples collected in the same study period with GeneXpert Ct > 27, agreement ranged from 31% to 100% (Table 1), indicating reduced to similar performance compared to GeneXpert.

To compare performance in terms of usability, machine size, storage requirements, and functionality, an overview of the characteristics of each platform and test is presented in Table 2. According to laboratory technicians' experience during the evaluation, the Galaxy Lite platform was less user-friendly than the other systems, requiring additional manual

**Table 1. Prospectively selected samples tested on Vivalytic, M10, Igenesis, and Flash10 platforms.**

Prospectively selected samples tested on Vivalytic, M10, Igenesis, Flash10 platforms

| | | Vivalytic | | | Flash10 | | | Galaxy lite | | | M10 1hr test | | | M10 1/2hr test | | |
|---|---|---|---|---|---|---|---|---|---|---|---|---|---|---|---|---|
| | | successfully tested / total n samples | positive (n) | % agreement to reference | successfully tested / total n samples | positive (n) | % agreement to reference | successfully tested / total n samples | positive (n) | % agreement to reference | successfully tested / total n samples | positive (n) | % agreement to reference | successfully tested / total n samples | positive (n) | % agreement to reference |
| Prospective samples collected based on GeneXpert results (6 samples positive for >1 target) | Positive for SARS-CoV-2 | 58/58 | 42 | 72.4% | 58/58 | 54 | 93.1% | 35/50 | 25 | 71.4% | 43/43 | 35 | 81.4% | 15/15 | 14 | 93.3% |
| | Negative for SARS-CoV-2 | 164/165 | 164 | 100.0% | 165/165 | 164 | 99.4% | 104/141 | 99 | 95.2% | 110/110 | 110 | 100.0% | 55/55 | 55 | 100.0% |
| | Positive for influenza A | 59/59 | 50 | 84.7% | 59/59 | 54 | 91.5% | 32/58 | 29 | 90.6% | 40/40 | 38 | 95.0% | 19/19 | 19 | 100.0% |
| | Negative for influenza A | 164/164 | 164 | 100.0% | 164/164 | 163 | 99.4% | 107/133 | 98 | 91.6% | 113/113 | 113 | 100.0% | 51/51 | 50 | 98.0% |
| | Positive for influenza B | 50/50 | 41 | 82.0% | 50/50 | 47 | 94.0% | 18/25 | 14 | 77.8% | 14/14 | 13 | 92.9% | 36/36 | 36 | 100.0% |
| | Negative for influenza B | 173/173 | 173 | 100.0% | 173/173 | 172 | 99.4% | 121/166 | 120 | 99.2% | 139/139 | 139 | 100.0% | 37/34 | 34 | 100.0% |
| | Positive for RSV | 62/62 | 58 | 93.5% | 62/62 | 62 | 100.0% | 57/62 | 51 | 89.5% | 59/59 | 59 | 100.0% | 3/3 | 3 | 100.0% |
| | Negative for RSV | 161/161 | 161 | 100.0% | 161/161 | 161 | 100.0% | 82/129 | 79 | 96.3% | 94/94 | 94 | 100.0% | 67/67 | 66 | 98.5% |
| Reference collection | Influenza A Reference strains | 19/20 | 18 | 94.7% | 20/20 | 19 | 95.0% | NA | NA | NA | NA | NA | NA | 20/20 | 20 | 100.0% |
| Reference collection | Influenza B Reference strains | 5/5 | 5 | 100.0% | 5/5 | 5 | 100.0% | NA | NA | NA | NA | NA | NA | 5/5 | 5 | 100.0% |
| Additional selection | Positive for influenza A with GeneXpert Ct value >27& <40 | 29/30 | 9 | 31.0% | 30/30 | 24 | 80.0% | 3/6 | 3 | 100.0% | 8/8 | 7 | 87.5% | 30/30 | 30 | 100.0% |

**Table 2. Practical details regarding the five PCR platforms evaluated in this study.**

| | GeneXpert | Vivalytic | FlashDetect Flash10 | Galaxy Lite | M10 |
|---|---|---|---|---|---|
| **Hardware** | | | | | |
| Size machine – length x width x height (in cm) | 27 x 30 x 31 (height is 47 cm with the TS unit on top) | 20 x 40 x 39 | 28 x 37 x 60 | 48 x 33 x 40 | M10 Console: 17 x 23 x 39 M10 Module: 14 x 33 x 32 |
| Number of slots per analyzer | 4 | 1 | 4 | 6 | 1 |
| Desk space machine (in cm) | 27x30 | 20 x 40 | 42.3 x 28.6 | 48 x 33 | 33 x 33 |
| Possibility to stack machines | Yes | No | No | No | Yes |
| Need for additional device | No | No | No | No | Console + Module |
| Desk space additional device | 0 | 0 | 0 | 0 | 16 x 200 |
| Random access | Random per slot | One slot per device | Random per slot | Batch of 6 | One slot per device |
| Results per hour (max) | 6.9 | 0.9 | 4 | 4.8 | 1.7 (30 min version) |
| Power supply cables | 1 cable, unit of 4 slots | 1 cable, per unit of 1 slot | 1 cable, unit of 4 slots | 1 cable, unit of 4 slots | 1 cable, units are linkable |
| Noise (dB) | <69 dB | ≤ 55 dB(A) in operating mode. Short term loudness can exceed mean loudness. | <58dB | Unknown | 26 dB in stand-by, < 50 dB during run |
| Scanner internal/external | Internal | Internal | Internal | External | Internal |
| Screen size (cm) | 10 inch | 7 inch (17,78 cm) | 15 inch (38.1 cm) | Unknown | 24,5x16cm (10,1 inch) |
| **Cartridges** | | | | | |
| Storage temperature | Room temperature | Room temperature | Room temperature | Fridge 4–8 C | Room temperature |
| Volume of 100 cartridges in boxes | 25.2 liters | 52 liters | 36 liters | Unknown | 38 liters |
| Firmness cartridges as assessed by the performing laboratory technician | Robust | Robust | Robust | Fragile | Robust |
| Amplification targets | SARS: RdRp-gene, E-gene, N2-gene Influenza A:Matrix-gene, PB2-gene, PA-gene Influenza B:NS-gene, Matrix-gene RSV:Nucleocapsid-gene for RSV A and RSV B Internal control: artificial target Sample Processing Control and Probe Check Control | SARS: N-gene, E-gene Influenza A:Matrix-gene Influenza B:NS2-gene RSV:Nucleocapsid-gene Internal control: artificial target Human control | SARS: Orf1ab-gene, N-gene Influenza A:Matrix-gene Influenza B:NS1-gene RSV:M-gene Internal control: RNase P gene | SARS: Orf1ab-gene, N-gene Influenza A:Matrix-gene Influenza B:NS1-gene RSV:F-gene Internal control: artificial target | SARS: Orf1ab-gene, E-gene, N-gene Influenza A:Matrix-gene, PB2-gene Influenza B:NS1-gene, M-gene RSV:M-gene, N-gene Internal control: artificial target |
| Brief description SOP | Open lid, insert sample, close lid, insert cartridge into the machine | Remove packaging, insert sample, close lid, insert cartridge into the machine | Open screw cap, insert sample, apply screw cap, remove protector from PCR chamber, insert cartridge into the machine | Remove seal, break away plastic appendages, remove empty fluid container, apply fluid container, remove lid of sample container, apply lid sample container, apply cartridge to cartridge slide, insert cartridge slide into the machine | Remove protective clip. Pierce cartridge, open lid, insert sample, close lid, insert cartridge into the machine |

*(Continued)*

**Table 2.** (Continued)

| | GeneXpert | Vivalytic | FlashDetect Flash10 | Galaxy Lite | M10 |
|---|---|---|---|---|---|
| **Sample** | | | | | |
| Volume | 300µl | 300µl | 600µl | 200µl | 300µl |
| Sample preparation | None | None | None | None | None |
| Sample medium | VTM, UTM, eNAT | UTM, eNAT | VTM or direct swab sample | Unknown | -Noble Bio CTM (Noble Biosciences: UTNFS-3B-2) -COPAN eNAT(Copan: 606CS01P) -COPAN Universal Transport Medium (recommended 3mLof UTM-RT medium) |
| **Handling time** | | | | | |
| Random access | Yes | Yes | Yes | No | Yes |
| Time to result | 30 minutes (as soon as 20 minutes with Early Assay Termination (EAT) for CoV-2 positives | 65 minutes | 35 minutes | 75 minutes ideally but longer because of limited availability related to batch analysis | 60 minutes/ 30 minutes |
| **Software** | | | | | |
| Sample tracking | Link between sample and cartridge made by scanning both items thereafter full sample tracking | Link between sample and cartridge made by scanning both items thereafter full sample tracking | Link between sample and cartridge made by scanning both items thereafter full sample tracking | Link between sample and cartridge made by scanning both items link of result to sample dependent on location of sample in analyzer with possibility of exchange of results | Link between sample and cartridge made by scanning both items thereafter full sample tracking |
| Ease of handling interface as assessed by the performing laboratory technician | Good, but results only visible in grid format | Good visibility Ct value and curve | Good visibility sample ID, Ct value and curve readably available on secondary screen | Good but hampered by small screen size | Good sample ID and result visible on primary screen Ct value and curve readably available on secondary screen, slightly hampered by small screen size |
| **Link to laboratory information system** | | | | | |
| HL7 | Yes, + HL7 and POCT1-A2 | Yes | Yes | Yes | Yes |
| Transfer of CT value | Yes | Yes | Yes | Yes | Yes |
| Automated transfer to Lab system without technician handling | Yes | Yes | Yes | Yes | Yes |
| **Point of Care feasibility** | | | | | |
| User accounts possible | Yes | Yes | Yes | Yes | Yes |
| Method of user registration | Username + password or barcodes | Barcode, DMC, user & password | Username + password | Username + password | Username + password |
| Automatic log off, time in minutes | Yes, customized between 15 and 500 minutes | Yes, 5 minutes | Yes, 30 mins | Yes, can be set in settings | No |
| User identification data exportable (who performed which test) | Exportable in CSV format | User identification (logging attempts & time) may be exported | Yes, exportable in pdf report and csv format. | Yes | Yes |

*(Continued)*

**Table 2.** (Continued)

| | GeneXpert | Vivalytic | FlashDetect Flash10 | Galaxy Lite | M10 |
|---|---|---|---|---|---|
| Results visibility | Per user, administrator | Per user, administrator | Per user, administrator | Per user | Per user, administrator |
| Overall ease of use as POCT | High | High | High | Low (handling is too complex and errone-ous for POCT in our experience) | Medium (because seal needs to be pierced which can cause malfunction if not per-formed correctly) |
| Costs | €€€€ | €€€ | €€ | €€ | €€ |

steps and being the only assay that could not be stored at room temperature. In the Dutch laboratory setting, all newly introduced platforms were less expensive per test than the GeneXpert platform, although such price differences may vary across countries and healthcare systems.

## Discussion

The use of rapid PCR platforms for respiratory virus detection has expanded significantly since the COVID-19 pandemic, driven by the need to improve patient flow, reduce patient isolation durations, and support efficient patient triage in emergency departments. Concurrently, many new platforms have entered the market, often promising faster turnaround times and reduced costs. This highlights the importance of comparative evaluations to guide evidence-based implementation.

In this study, we assessed four rapid PCR systems against GeneXpert as the reference test for the detection of SARS-CoV-2, IAV/IBV, and RSV on clinical specimens. Compared to the 60-minute version, the M10 30-minute assay version showed higher agreement, despite the limited number of samples tested. Flash10 also performed well, but with slightly lower agreement for SARS-CoV-2 and IAV for mainly samples with low viral load. Vivalytic showed more variable performance, with high agreement for RSV and negative samples, but lower agreement for SARS-CoV-2 (72.4%) and IAV (84.7%). However, it should be mentioned that according to the manufacturer, the sample medium used (VTM) has not been validated for the test. Galaxy Lite was excluded midway due to persistent technical issues, including high error rates in two separate instruments. According to the manufacturer, these issues have since been resolved. However, it was too late to be able to include the modified test in our comparative analysis. Our findings for GeneXpert and M10 (60 minutes) align with previously published studies evaluating similar assays [4,5,7,8,10,11]. The lower performance of Vivalytic for SARS-CoV-2 and IAV in our evaluation is also consistent with an earlier report describing reduced agreement, particularly in samples with low viral loads [10].

With the exception of M10, all platforms showed a modest reduction in agreement for low viral load samples, particularly evident in IAV cases with GeneXpert Ct values above 30. This difference was intentionally further explored through the inclusion of additional samples with high Ct-values. Although relevant from an analytical perspective, this reduction may be of limited clinical impact in emergency settings, where rapid turnaround and decision-making are prioritized over detection of very low viral loads with uncertain clinical relevance [12].

A critical factor contributing to platform-specific performance is the design of the assays, particularly the number of PCR targets and different target genes used per virus (Table 2). All platforms use two or more targets for SARS-CoV-2 (e.g., E gene and N2 or ORF1ab). This on purpose redundancy offers a benefit in case of target dropout due to viral mutation. It also can increase the analytical sensitivity in clinical specimens, especially if the PCR amplifies multi copy RNAs, like mRNA or intermediate viral RNA copies. However, with the exception of GeneXpert and M10 (Table 2), the remaining platforms use only a single target for RSV and for both IAV and IBV. While this simplifies assay design, it introduces a vulnerability to false-negative results if mutations occur in the targeted region. This risk is especially relevant for genetically

 

dynamic viruses such as SARS-CoV-2 and has also been described for IAV, where mutations in the matrix or hemagglutinin gene regions have occasionally led to reduced assay performance or target failure in single-target designs [13,14]. The inclusion of genetically diverse IAV and IBV strains strengthens the analytical evaluation of the platforms and reflects broader diagnostic challenges, including preparedness for zoonotic variants as emphasized in recent national EQA studies [13].

Strengths of this study include its prospective design, parallel testing of identical clinical specimens across platforms, and inclusion of both routine and high-Ct IAV samples as well as cultured influenza virus strains (from different genetic clades and H- and N-subtypes including potential zoonotic viruses). Limitations include the absence of blinding, pre-defined group sizes, and limited sample numbers for certain platforms (e.g., SD Biosensor with two different kits) or subgroups of respiratory viruses. A separate group of samples negative for all viral targets was not included in this study. However, for each pathogen, negative results were inherently represented by samples positive for other viral targets, ensuring that each assay's specificity could still be evaluated within the tested cohort. Nevertheless, the absence of a distinct all-negative control group may limit a full assessment of assay performance in populations with low pathogen prevalence.While the inclusion of genetically diverse IAV and IBV strains strengthens the analytical evaluation of the platforms, the continual emergence of viral variants poses an ongoing challenge to assay robustness. Under the EU In Vitro Diagnostic Regulation (IVDR), manufacturers of high-risk diagnostic tests are required to demonstrate sustained performance, including in the face of viral evolution. This process involves extensive documentation, validation, and in some cases collaboration with EU Reference Laboratories (EURLs), which may delay rapid assay adaptation. Ultimately, platform resilience depends on the manufacturer's ability to proactively update assay design and meet evolving regulatory demands in a timely and efficient manner. This is particularly relevant for newly introduced platforms, which often rely on single-target assay designs and have not yet demonstrated long-term performance across diverse clinical settings. Ongoing independent evaluations and transparent dissemination of performance data will be essential to ensuring sustained diagnostic reliability.

In conclusion, several of the rapid PCR platforms evaluated in this study show strong diagnostic potential and may serve as reliable alternatives in clinical workflows using standard laboratory-based NAAT, provided platform-specific limitations are considered. Future studies should continue to assess not only analytical performance but also test stability, workflow integration, and resilience to viral genetic variability.

## Acknowledgments

We would like to thank Gabriel Goderski, Sharon van de Brink, Samantha Zoomer, John Sluimer (NIC location RIVM, Bilthoven), Ron Fouchier, Oanh Vuong (NIC location Erasmus MC, Rotterdam), Erhard van der Vries, and Manon Houben (Royal GD, Deventer) for sharing the cultured IAV and IBV viruses.

## Author contributions

**Conceptualization:** Pieter W. Smit, Peter de Man.

**Data curation:** Henk Brand, Simone Breijer, Mariska van Wijk.

**Formal analysis:** Pieter W. Smit, Adam Meijer.

**Funding acquisition:** David S. Y. Ong.

**Investigation:** Mariska van Wijk, David S. Y. Ong.

**Methodology:** Peter de Man, Henk Brand, Simone Breijer, Mariska van Wijk, Adam Meijer.

**Project administration:** Henk Brand, Simone Breijer, Mariska van Wijk.

**Supervision:** Pieter W. Smit, Adam Meijer, David S. Y. Ong.

**Validation:** Pieter W. Smit, Simone Breijer.

**Visualization:** Pieter W. Smit.

**Writing – original draft:** Pieter W. Smit, Peter de Man, Adam Meijer, David S. Y. Ong.

**Writing – review & editing:** Pieter W. Smit, Henk Brand, Simone Breijer, Mariska van Wijk.

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
