## [Decision Letter · Decision Letter 0]

28 Oct 2025

Dear Dr.  Smit,

We look forward to receiving your revised manuscript.

Kind regards,

Minal Dakhave

Academic Editor

PLOS ONE

Journal Requirements:

2. We note that your Data Availability Statement is currently as follows: “All relevant data are within the manuscript and its Supporting Information files”

Additional Editor Comments:

The manuscript presents an informative comparative evaluation of diagnostic performance across various platforms. However, inclusion of complete negative sample data is essential to enable a comprehensive specificity analysis and strengthen the overall conclusions. The primary objective—to assess diagnostic accuracy, usability, and operational costs—has been stated but is not clearly reflected in the Results or Discussion sections. The current focus remains largely on positive samples; therefore, inclusion of negative sample testing and elaboration on usability and cost parameters would provide a more balanced interpretation.

For samples that tested positive on one platform but were undetected on others, it is recommended to repeat testing at least once to confirm negative results, thereby improving data reliability. Additionally, the rationale and methodology for sample size calculation should be included to enhance transparency and reproducibility. In the Results section, a more detailed statistical analysis is advised to improve clarity and interpretability of findings.

The authors are also requested to carefully address all reviewer comments and revise the manuscript in accordance with the journal’s publication guidelines.

Reviewer's Responses to Questions

**Comments to the Author**

1. Is the manuscript technically sound, and do the data support the conclusions?

Reviewer #1: Yes

2. Has the statistical analysis been performed appropriately and rigorously?

Reviewer #1: Yes

3. Have the authors made all data underlying the findings in their manuscript fully available?

Reviewer #1: Yes

4. Is the manuscript presented in an intelligible fashion and written in standard English?

Reviewer #1: Yes

Reviewer #1: This study offers valuable comparative data on emerging rapid PCR platforms for respiratory virus detection. The manuscript is well-organized and clearly written, but several points need clarification.

The study aims to compare diagnostic performance against GeneXpert but only includes positive samples, which limits the assessment of specificity and predictive values. The authors should clearly state in the Methods section that only GeneXpert-positive samples were included and discuss this limitation in the Discussion. Additionally, the rationale for the number of samples per virus should be clearly explained.

The description of the statistical analysis is brief and should specify how agreement percentages were calculated, such as whether based on simple proportion, Cohen’s kappa, or diagnostic accuracy metrics like sensitivity and specificity. It should also clarify how invalid or failed runs were handled in the analysis, whether they were excluded or counted as test failures, especially since test failures were only reported for the Galaxy Lite platform (27.2%), while the other systems nearly always produced valid results. This clarification would improve methodological transparency.

The results are generally appropriate for the study’s comparative aims and show clear differences in performance between platforms. However, because only GeneXpert-positive samples were included and no true negative group was tested, the study cannot evaluate specificity or predictive values. While reporting agreement percentages is suitable for comparative evaluation, this limitation should be explicitly acknowledged in the Discussion to prevent overinterpreting the diagnostic accuracy. Additionally, the small sample size for some subgroups and the exclusion of Galaxy Lite due to high failure rates should also be acknowledged.

The findings are acceptable and informative for preliminary evaluation but should be interpreted cautiously in the absence of a full positive-negative dataset.

**Do you want your identity to be public for this peer review?** For information about this choice, including consent withdrawal, please see our Privacy Policy

Reviewer #1: **Yes: ** Lawrence Annison

---

## [Author Response · Author response to Decision Letter 1]

25 Nov 2025

Dear Editor and Reviewer,

Thank you for the careful and constructive review of our manuscript (Title: Comparative Evaluation of Five Rapid PCR Platforms for Respiratory Virus Detection, Manuscript ID: PONE-D-25-41222). We appreciate the reviewers’ time and comments. We have prepared a point-by-point response (attached) and made the following key revisions to the manuscript:

• Emphasis on negative samples: We clarified and strengthened the presentation regarding the absence of negative samples and discussed its implications throughout the Methods and Results sections.

• Statistical analysis: We added appropriate statistical analyses to support our findings (see Methods).

• Study design and limitations: We expanded the Discussion to explicitly acknowledge limitations of the study design, including potential biases and constraints that affect interpretation and generalizability.

• Costs and user operability: The Results section now places greater emphasis on cost considerations and practical aspects of user operability, with additional text quantifying or qualifying these factors.

• Additionally, when preparing the raw data file, we cross-checked again all data and decided to represent the data slightly differently for easier representation in combination with the raw data. This had marginal effect on the total number of negative samples tested and no effect on our findings. To reflect this update, we have corrected Table1.

All reviewer comments have been addressed point-by-point in this document. Where we did not follow a suggested change, we provide a brief justification.

We believe these revisions improve the clarity and robustness of the manuscript and welcome further feedback.

Sincerely,

Dr. P. Smit

Dr. D.S.Y. Ong

Point by point rebuttal below

RESPONSE

We have checked the PLOS ONE’s style requirements and adopted accordingly.

2. We note that your Data Availability Statement is currently as follows: “All relevant data are within the manuscript and its Supporting Information files”

RESPONSE

We appreciate the effort of Plos One in making underlying data more accessible and transparent. In that regard, we have added the raw data in a supplementary table 1.

RESPONSE

We deleted the sentence that referred to the data not shown comment in the discussion as requested, because it did not belong to the core part of the research.

RESPONSE

We thank the editor for this remark. The reviewer has not given a recommendation to cite specific published work.

RESPONSE

Thank you, we have checked the reference list and we do not have retracted cited papers.

Additional Editor Comments:

The manuscript presents an informative comparative evaluation of diagnostic performance across various platforms. However, inclusion of complete negative sample data is essential to enable a comprehensive specificity analysis and strengthen the overall conclusions.

RESPONSE

We agree with the editor that a full prospective study that would also consecutively include all negative samples would have been preferred. However, with an expected prevalence for each of the different viruses, this would have dramatically increased the sample size to end up with a sufficient number of positive samples or it would have led to lower number of positive samples, thereby limiting the accurateness of the sensitivity assessment. However, to emphasize this limitation, we included a sentence in the discussion:

“A separate group of samples negative for all viral targets was not included in this study. However, for each pathogen, negative results were inherently represented by samples positive for other viral targets, ensuring that each assay’s specificity could still be evaluated within the tested cohort. Nevertheless, the absence of a distinct all-negative control group may limit a full assessment of assay performance in populations with low pathogen prevalence. “

The primary objective—to assess diagnostic accuracy, usability, and operational costs—has been stated but is not clearly reflected in the Results or Discussion sections. The current focus remains largely on positive samples; therefore, inclusion of negative sample testing and elaboration on usability and cost parameters would provide a more balanced interpretation.

RESPONSE

We agree with the editor that our comment on costs and usability was very short. We have added the following to the result section:

“According to laboratory technicians’ experience during the evaluation, the Galaxy Lite platform was less user-friendly than the other systems, requiring additional manual steps and being the only assay that could not be stored at room temperature. In the Dutch laboratory setting, all newly introduced platforms were less expensive per test than the GeneXpert platform, although such price differences may vary across countries and healthcare systems.”

For samples that tested positive on one platform but were undetected on others, it is recommended to repeat testing at least once to confirm negative results, thereby improving data reliability.

RESPONSE

We agree with the reviewer from a scientific point of view that this would have been of added value. In this study we wanted to mimick as closely as possible to the clinical use case of these rapid diagnostic tests and decided to do this prospectively in a routine hospital laboratory setting. In almost all clinical settings, a test result would not be repeated if negative. Therefore, we kept the result as they were. However, in case a test failed, we repeated the sample. This would also represent real-world clinical practice as otherwise no test result could be reported to the clinicians.

We have added a note on this in the method section:

“To represent most clinical settings, invalid test results were repeated, but discrepant test results in comparison to other platforms were not repeated.”

Additionally, the rationale and methodology for sample size calculation should be included to enhance transparency and reproducibility. In the Results section, a more detailed statistical analysis is advised to improve clarity and interpretability of findings.

The authors are also requested to carefully address all reviewer comments and revise the manuscript in accordance with the journal’s publication guidelines.

RESPONSE

We value the suggestions from the editor and added a sample size calculation.

“Sample size calculation was performed with the aim to be able to detect at least a 15% difference in agreement per viral target with 80% power and an α of 0.05 (McNemar), which showed that 53 samples were required per viral target.”

Regarding the question of the editor to add statistical analyses, we kindly disagree that adding statistics to this relatively plain and straightforward manuscript would be of any added value. In our opinion we do not make a strong statement that needs to be supported by any statistics and therefore see no added value.

Reviewer's Responses to Questions

Comments to the Author

Reviewer #1: This study offers valuable comparative data on emerging rapid PCR platforms for respiratory virus detection. The manuscript is well-organized and clearly written, but several points need clarification.

The study aims to compare diagnostic performance against GeneXpert but only includes positive samples, which limits the assessment of specificity and predictive values. The authors should clearly state in the Methods section that only GeneXpert-positive samples were included and discuss this limitation in the Discussion. Additionally, the rationale for the number of samples per virus should be clearly explained.

RESPONSE

We thank the reviewer for the kind words and the suggestion to describe more clearly this limitation in the method and discussion sections. The discussion section was already adjusted based on the editors comments , see comment above. And below:

“A separate group of samples negative for all viral targets was not included in this study. However, for each pathogen, negative results were inherently represented by samples positive for other viral targets, ensuring that each assay’s specificity could still be evaluated within the tested cohort. Nevertheless, the absence of a distinct all-negative control group may limit a full assessment of assay performance in populations with low pathogen prevalence. “

Based on this comment we adusted the method section. It now reads:

“Positive samples according to GeneXpert were enrolled consecutively until the target number per virus was reached. No separate group of negative samples was included.”

The description of the statistical analysis is brief and should specify how agreement percentages were calculated, such as whether based on simple proportion, Cohen’s kappa, or diagnostic accuracy metrics like sensitivity and specificity. It should also clarify how invalid or failed runs were handled in the analysis, whether they were excluded or counted as test failures, especially since test failures were only reported for the Galaxy Lite platform (27.2%), while the other systems nearly always produced valid results. This clarification would improve methodological transparency.

RESPONSE

Thank you for these suggestions, we have included a statement regarding the invalid test results based on editors suggestion (see above) and invalid test results are also shown in table 1.

The method section now reads:

“Key performance metrics such as concordance and percentage agreement ratios, including qualitative assessments of user experiences and operational reliability, were assessed for each platform”

The results are generally appropriate for the study’s comparative aims and show clear differences in performance between platforms. However, because only GeneXpert-positive samples were included and no true negative group was tested, the study cannot evaluate specificity or predictive values. While reporting agreement percentages is suitable for comparative evaluation, this limitation should be explicitly acknowledged in the Discussion to prevent overinterpreting the diagnostic accuracy. Additionally, the small sample size for some subgroups and the exclusion of Galaxy Lite due to high failure rates should also be acknowledged.

The findings are acceptable and informative for preliminary evaluation but should be interpreted cautiously in the absence of a full positive-negative dataset.

RESPONSE

Thank you for this suggestion which fits with the suggestions of the editor. We have made this limitation more explicit in the discussion. It now reads:

“Limitations include the absence of blinding, predefined group sizes, and limited sample numbers for certain platforms (e.g. SD Biosensor with two different kits) or subgroups of respiratory viruses. A separate group of samples negative for all viral targets was not included in this study. However, for each pathogen, negative results were inherently represented by samples positive for other viral targets, ensuring that each assay’s specificity could still be evaluated within the tested cohort. Nevertheless, the absence of a distinct all-negative control group may limit a full assessment of assay performance in populations with low pathogen prevalence. “

---

## [Editor Report · Decision Letter 1]

26 Nov 2025

Comparative Evaluation of Five Rapid PCR Platforms for Respiratory Virus Detection

PONE-D-25-41222R1

Dear Dr. Pieter Willem Smit,

We’re pleased to inform you that your revised manuscript has been judged scientifically suitable for publication and will be formally accepted for publication once it meets all outstanding technical requirements.

Kind regards,

Dr. Minal Dakhave

Academic Editor

PLOS ONE

---

## [Editor Report · Acceptance letter]

PONE-D-25-41222R1

PLOS ONE

Dear Dr. Smit,

I'm pleased to inform you that your manuscript has been deemed suitable for publication in PLOS ONE. Congratulations! Your manuscript is now being handed over to our production team.

Kind regards,

on behalf of

Dr. Minal Dakhave

Academic Editor

PLOS ONE